# Inhibitory Effects of *Scutellaria baicalensis* Root Extract on Linoleic Acid Hydroperoxide-induced Lung Mitochondrial Lipid Peroxidation and Antioxidant Activities

**DOI:** 10.3390/molecules24112143

**Published:** 2019-06-06

**Authors:** Pei Ru Liau, Ming Shun Wu, Ching Kuo Lee

**Affiliations:** 1School of Pharmacy, College of Pharmacy, Taipei Medical University, Taipei 11031, Taiwan; d301099009@tmu.edu.tw; 2Division of Gastroenterology, Department of Internal Medicine, Wan Fang Hospital, Taipei 11661, Taiwan; 3School of Medicine, College of Medicine, Taipei Medical University, Taipei 11031, Taiwan

**Keywords:** *Scutellaria baicalensis*, phytochemical, rat lung, lipid peroxidation, antioxidant

## Abstract

In this study, we evaluated the ability of *Scutellaria baicalensis* Georgi to protect lipid-peroxidation (LPO) in lung tissue after free radical-induced injury. We prepared *S. baicalensis* root (SBR) extracts using different solvents. The total flavonoid and total phenol contents of each extract were measured, and the ROS damage protection was evaluated by analyzing linoleic acid hydroperoxide (LHP)-induced LPO in rat lung mitochondria. Moreover, evaluating diphenylpicrylhydrazyl (DPPH), hydrogen peroxide, superoxide anion radical, and hydroxyl radical scavenging abilities and using metal chelating assays were used to determine in vitro antioxidant activity. The ethyl acetate (EtOAc) extract showed high ROS scavenging ability, and four compounds were subsequently isolated and purified from this extract: baicalin, baicalein, wogonin, and oroxylin A. Baicalein in rat lung mitochondria the most significant LHP-induced LPO inhibition was shown and extracted with EtOAc that contained the highest amount of baicalein. Thus, baicalein and the EtOAc extract of SBR may be efficient in conferring ROS damage protection and inhibiting LHP-induced LPO in rat lung mitochondria. Additional studies are warranted to investigate their use as antioxidant therapy for respiration infections, nutrition supplements, and lead compounds in pharmaceuticals.

## 1. Introduction

Numerous epidemiological investigations have highlighted the association between air pollution and respiratory diseases in recent decades, n; for example, pollutants exposure can cause reduction of lung function, chronic obstructive pulmonary disease (COPD), asthma, and cough [1]. The number of deaths attributable to ambient air pollution exposure has been estimated to be 3.7–4.8 million, representing 7.6% of the total global mortality, according to a study that was conducted by Cohen et al. [2]. In previous investigated, nanoparticles caused oxidative stress, genotoxicity, and inflammatory responses, as seen by the significant induction of ROS, LPO, micronucleus (MN), tumor necrosis factor-α (TNF-α), interleukin-1β (IL-1β), and interleukin-6 (IL-6) [3]. According to World Health Organization (WHO), 29% of all deaths and deaths from lung cancer and 43% of all deaths and deaths from COPD are attributed to atmospheric particulate matter exposure and they are independent of cigarette smoking [4]. In 2014, >6000 deaths were attributed to ischemic heart disease, stroke, lung cancer, and COPD in Taiwan [5]. Atmospheric particulate matter induced epigenetic changes, including microRNA dysregulation, DNA methylation, microenvironment alteration, and cell autophagy and apoptosis, can result in oncogene activation and tumor suppressor gene inactivation in lung cancer [6]. Numerous studies have suggested whether oxidative stress is caused by the overproduction of various free radicals. Free radicals accumulate when there is an imbalance between antioxidants and oxidants, causing serious damage to macromolecules, such as nucleic acids, proteins, and lipids [7]. Moreover, the imbalance is a major risk factor for lung dysfunction [8]; the oxidative exacerbation rate can be effectively reduced through antioxidant therapy. Therefore, it is important to investigate substances with antioxidant activity than can protect lung tissues and reduce lung injury that is caused by oxidative stress. *Scutellariae* Radix is the dried root of *Scutellaria baicalensis* Georgi, which is a species of flowering plant that belongs to the *Labiatae* family; the dried root is a widely used “heat-clearing” herb in Traditional Chinese Medicine. Modern researches have revealed that the root has anticancer [9], antiviral [10], antibacterial [11], anti-inflammatory [12], and other pharmacological activities; it also has favorable antioxidant activity [13,14]. In this study, we evaluated the ability of *S. baicalensis* root (SBR) extracts to protect the lungs from damage that is caused by LPO and free radicals.

## 2. Results

### 2.1. Total Flavonoid and Total Phenol Contents of SBR Extracts

Many “heat-clearing” herbs derive their antioxidative activity from phenolic compounds [15]. Therefore, we estimated the total flavonoid and total phenol contents of SBR extracts. Our results showed that the EtOAc extract contained the highest total phenol and total flavonoid contents, which were approximately 3- and 2.5-times higher than the contents in the water extract, respectively.

### 2.2. Free Radical Scavenging and Antioxidant Activities

SBR was extracted with water, ethanol (EtOH), acetone, and EtOAc, respectively, and it obtained the four solvent extracts. After removing the residue, the filtrates were combined and concentrated while using an evaporator to obtain four extracts of SBR. All of the extracts were stored at −20 °C in a refrigerator for subsequent experiments. Table 1 lists data pertaining to each SBR extract, which presents the effects of SBR extracts on free radical scavenging and antioxidant activities.

#### 2.2.1. DPPH Radical Scavenging Activity

DPPH is widely used to estimate the free radical scavenging activity of various antioxidants. In this study, the free radical scavenging activity of each SBR extract and gallic acid, which was used as the positive control, was determined while using the DPPH-based method. The obtained results indicated that the EtOAc extract showed higher scavenging activity (IC_50_ = 13.08 µg/mL) than other solvent extracts, and is therefore worthy of further investigation.

#### 2.2.2. Antioxidant Activity of SBR Extracts and Their Superoxide (O_2_•−), Hydroxyl (−OH) and Hydroperoxide (H_2_O_2_) Scavenging Activities

When compared with other solvent extracts, the EtOAc extract showed the most efficient antioxidant activity (Table 1). The −OH and H_2_O_2_ scavenging activities that were displayed by the four solvent extracts did not significantly differ. Moreover, the antioxidant and −OH and H_2_O_2_ scavenging activities were not-directly associated with total flavonoid and phenol content.

#### 2.2.3. Ferrous Ion-chelating Activity

Among the transition metals, when considering its high reactivity, iron is the most important prooxidant of lipid oxidation. The ferrous state of iron accelerates lipid oxidation by degrading hydrogen and lipid peroxides into reactive free radicals via the Fenton reaction: Fe^2+^ + H_2_O_2_ → Fe^3+^ + OH− + •OH. Bivalent ferrous ions (Fe^2+^) are the most powerful prooxidants among the various species of metal ions. In this study, the chelating activities of the water extract and EDTA (positive control) were determined by ferrous ion-chelating activity, with IC_50_ values of 184.82 ± 3.21 µg/mL and 8.17 ± 0.01 µg/mL, respectively, and the IC_50_ values with the other extract were >200 µg/mL. The results indicated that the high ferrous ion-chelating activity is probably related to the hydrophilic component of the water extract.

### 2.3. Inhibitory Effects of SBR Extracts on LHP-induced LPO

In this study, the inhibitory effects of SBR extracts and Trolox (positive control) on LHP-induced LPO were determined (Table 2). Table 2 presents the IC_50_ values of the SBR extracts and Trolox. The inhibitory effects of SBR extracts followed this order: EtOAc > acetone > EtOH ≥ water.

### 2.4. Phytochemicals Isolated from High-potential Extract and Structure Identification

Based on the aforementioned data, the EtOAc extract showed high ROS scavenging ability, and four compounds were isolated and purified from the EtOAc extract, and their structure was subsequently determined:

Compound I: Yellow crystals using ESI-MS and NMR for identification. The mass spectrum exhibited a [M + H]^+^ peak at an *m*/*z* of 447, and the ^1^H-NMR spectrum in CD_3_OD was the same as that of baicalin reported in the literature [16]. Thus, we identified compound I as baicalin.

Compound II: Yellow powder. The mass spectrum exhibited a [M + H]^+^ peak at an *m*/*z* of 271, and the ^1^H-NMR spectrum in CD_3_OD showed the same as those of baicalein by Huang et al. [17].

Compound III: Yellow powder. The mass spectrum exhibited a [M + H]^+^ peak at an *m*/*z* of 285, and the ^1^H-NMR spectrum in DMSO-d6 are those of wogonin by Huang et al. [17]. Thus, we identified compound III as wogonin.

Compound IV: Bright yellow crystals. The mass spectrum exhibited a [M + H]^+^ peak at an *m*/*z* of 285, and the ^1^H-NMR spectrum in DMSO-d6 are as those of oroxylin A by Trang et al. [18]. Thus, we identified compound IV as oroxylin A.

The four compounds belong to the flavones group, and Figure 1 shows their structures.

### 2.5. LPO Inhibition, free Radical Scavenging and Antioxidant Activities by Main Components of the EtOAc Extract

Baicalin (**1**), baicalein (**2**), wogonin (**3**), and oroxylin A (**4**) were isolated and purified from the EtOAc extract, and their LPO inhibitory were determined (Table 3). The inhibitory effects of these four compounds followed this order: baicalein > baicalin > wogonin > oroxylin A. Therefore, the inhibition of lipid peroxidative principle constituent is baicalein. Table 4 lists the free radical scavenging and the antioxidant activities of baicalin, baicalein, wogonin, and oroxylin A. In summary, the effect on antioxidant activity of baicalein is stronger than baicalin, wogonin and oroxylin A, we calculated the IC_50_ concentrations of DPPH, O_2_•− scavenging activity and Fe^2+^ chelating activity of baicalein as 2.80 ± 0.05, 43.99 ± 1.66 and 2.38 ± 0.69 µg/mL, respectively.

### 2.6. Composition of Each SBR Extract

HPLC is a widely used separation technique [19]. Chromatographic methods are considered to be the best when considering that they are easy and offer selectivity, rapidity, and reproducibility. The compositions of the water, EtOH, acetone, and EtOAc extracts and baicalin, baicalein, wogonin, and oroxylin A were analyzed using HPLC (Figure 2 and Figure 3). As stated earlier, baicalein showed the most significant LHP-induced LPO inhibition in rat lung mitochondria; and, the EtOAc extract showed the highest amount of baicalein (248.05 ± 32.11 µg/mg). The amount of baicalein in the four SBR extracts followed this order: EtOAc > acetone > EtOH > water.

## 3. Discussion

In view of this result, we study the antioxidative capacity of SBR with various types of ROS, which can scavenge from SBR extracts with different solvent. ROS include O_2_•−, peroxyl (ROO•), alkoxy (RO•); and, −OH. Superoxide is the primary oxygen free radical that is produced in mitochondria via the slippage of an electron from the electron transport chain to molecular oxygen during oxidative phosphorylation. O_2_•− is also produced when the neutrophils are killing a wide range of microorganisms with nicotine adenine dinucleotide phosphate (NAD(P) H) oxidase to produce O_2_•−. H_2_O_2_ would be catalyzed by enzyme to produce hydroxyl free radical −OH, which is very reactive and rapidly attack the molecules in nearby cells, regulated by metal ion, and is produced from hydroperoxide catalyzed by ferrous ion. Peroxyl causes lipid peroxidation, and hydroperoxide is less reactive and can be deleted by catalase when unnecessary. It is accepted that ROS play different roles *in vivo*. However, ROS may be very damaging, since they can attack lipids in cell membranes, proteins in tissues or enzymes, carbohydrates, and DNA, to induce oxidations, and it is considered to play a causative role in aging and several degenerative diseases that are associated with cancers, cell injury, inflammation, intestinal diseases, and several other debilitating diseases. Hence, it was suggested that the consumption of antioxidant-rich diets would help to alleviate diseases that are caused by ROS [20]. 

Oxidant induced abstraction of a hydrogen atom from an unsaturated fatty acyl chain of membrane lipids initiates the process of LPO, which propagates as a chain reaction [21]. Phytochemical studies on SBR showed that flavonoids are the major bioactive components [22,23]. Flavonoids have been identified as a kind of important antioxidant, according Halliwell and Guttrtidge [24] definition, any substance that delays, prevents, or remove oxidative damage to a target molecule, and, as a general principle of antioxidant defense, flavonoids have been identified as fulfilling most of the antioxidant criteria described, and they include: (1) suppressing ROS formation either by chelating transition metals or the inhibition of enzymes that are involved in free radical production and scavenging ROS; and, (2) upregulating or protecting antioxidant defenses. In our study, baicalein, wogonin, or baicalin were reported to inhibit xanthine oxidase [25,26], lipoxygenase [27], cyclooxygenase [28], microsomal monooxygenase, and glutathione S-transferase [29,30,31], and they are all involved in ROS generation. 

Some trace metal ions are essential for many physiological functions, as constituents of hemoproteins and cofactors of different enzymes in the antioxidant defense, including iron for catalase, copper for ceruloplasmin and Cu, Zn-superoxide dismutase, but modern life overuse of the nutritional supplement would increase the risk of free radical produce. Free iron is a potential enhancer of ROS formation, as exemplified by the reduction of hydrogen peroxide with the generation of the highly aggressive hydroxyl radical, and we obtained baicalin to chelate Fe^2+^ efficiently, which is consistent with the SBR water extract results. Baicalin may play an important role in oxygen metabolism for iron chelating activity.

When compared with the main compound of each extract, lower content aglycones (baicalein, oroxylin A, wogonin) significantly inhibited LHP-induced lipid peroxidation. Furthermore, EtOAc extract in which baicalein were richer than the other extracts and showed higher activity of inhibited LHP-induced LPO than the other extracts. LPO reaction of polyunsaturated fatty acids forms malondialdehyde, which causes cell damage, leading to a variety of pathological conditions, especially acute respiratory distress syndrome [32]. In these conditions, in response to various inflammatory stimuli, lung endothelial cells, alveolar cells, alveolar macrophages, and polymorphonuclear leucocytes release free radicals and ROS, resulting in extensive tissue destruction. In rats, lung microsomes have been reported to peroxidize at a 25–50-fold lower rate than the liver, kidney, testes, and brain microsomes, revealing the unique resistance of the lung to LPO. This can be attributed to the ratio of vitamin E to peroxidizable polyunsaturated fatty acids in lung microsomes; the ratio is several-fold higher in lung microsomes than in microsomes from other tissues [33]. In our study, we examined the inhibition of lipid peroxidation in rat lung mitochondria using LHP as inducer, which is closer to the real conditions of lung in vivo. 

A study that compared the structure–activity relationship between baicalein and wogonin reported that the quenching degree, spontaneous binding reaction through electrostatic interactions, and conformational changes might all be associated with the same skeleton structure of flavones. Based on experimental data of the differences in free radical scavenging and antioxidant activities that may be associated with the different substituent connected at ring-B, baicalein has more hydroxyl connected, and a larger conjugated system causes the higher biological activity, but the introduction of methoxyl at C-8 of wogonin increased the steric hindrance going against biological activity [34]. This may be the reason for the lowered scavenging activities displayed by wogonin and oroxylin A.

SBR is a traditional Chinese medicine that is known as Huang-Qin, it was the first recorded in Shennong Bencaojing (The Classic of Herbal Medicine), and written between about 200 and 250 AD, for the treatment of bitter, cold, lung, and liver problems [35]. Recent studies have reported the anti-inflammatory properties activities to treat lung injury of SBR, the ability to blocked protein expression of inducible NO synthase (iNOS), cyclooxygenase-2 (COX-2), phosphorylation of IκB-α protein and MAPKs in LPS-induce lung injury [36], the therapeutic effects of flavonoids-enriched extract from SBR on acute lung injury (ALI) induced by influenza A virus (IAV) in mice [37], reduced tumor progression and metastasis, directly induced tumor cell death, and inhibited tumor angiogenesis [38], attenuated the lung injury that was induced by myocardial ischemia and reperfusion, and, combined with cisplatin decreased the secretion of tumor necrosis factor-α of A549 cells, indicated a promising alternative method for lung cancer [39]. Accordingly, the present data (although far from conclusive) evidence a possible lung damage protection role of baicalein and the EtOAc extract of SBR.

## 4. Materials and Methods 

### 4.1. Extraction and Sample Preparation

The dried of *Scutellaria baicalensis* George was purchased from crude drug market in Taipei, Taiwan. Dry SBR (2 kg) was thoroughly washed under running tap water to remove any impurities, cut into small pieces, and then dried in an air circulating oven at 40 °C for 48 h. Each dried SBR (100 g) was extracted twice with four different one-liter solvents (water, ethanol (EtOH), acetone, and ethyl acetate (EtOAc)). The filtrates were combined and a concentrated evaporator was used to obtain four SBR extracts. All of the extracts were stored in a refrigerator for subsequent experiments.

### 4.2. Chemicals and Reagents

Folin–Ciocalteu reagent, hydrogen peroxide (H_2_O_2_), iron (III) chloride anhydrous, iron (III) chloride hexahydrate, linoleic acid, lipoxidase, nitroblue tetrazolium (NBT), horseradish peroxidase (HRPase), superoxide dismutase (SOD), phenazine methosulfate, phenol red, trichloroacetic acid (TCA), thiobarbituric acid (TBA,) and other chemicals and analytical grade solvents were purchased from Sigma Chemical Co (St. Louis, MO, USA).

### 4.3. Experimental Animals

Male Wistar rats were purchased from the Center of Experimental Animals, National Taiwan University, Taipei, Taiwan. All of the experiments were performed in accordance with guidelines for animal experiments of Taipei Medical University and Guide for the Care and Use of Laboratory Animals approved by the Chinese Society of Laboratory Animal Sciences, Taiwan. All efforts were made to minimize animal suffering and reduce the number of animals used.

### 4.4. Free Radical Scavenging and Antioxidantactivity Measurement

#### 4.4.1. Total Phenol Content

The total phenol content was determined using the Folin–Ciocalteu method [40], with some modifications. The stock solutions of SBR extracts (100 µL) were mixed with 500 µL of Folin–Ciocalteu reagent (diluted to 1:10 with distilled water), and 400 µL of 7.5% Na_2_CO_3_ was then added. After incubating this mixture at 50 °C in a water bath for 30 min, the absorbance was measured at 600 nm using an ELISA spectrophotometer (uQuant, Bio-Tek Instruments Inc., Winooski, VT, USA). Gallic acid monohydrate (1.5–200 µg/mL) was used as the standard to construct a calibration curve. The total phenol content is expressed as gallic acid equivalent (µg of GAE)/mg dry weight of the samples.

#### 4.4.2. Total Flavonoid Content

The stock solutions of SBR extracts (100 µL, 20 mg/mL) were transferred into a 1.5-mL microcentrifuge tube. Distilled water was added to obtain a volume of 0.3 mL, and 0.03 mL of NaNO_2_ (5%) was added 5 min later. Subsequently, 0.03 mL of AlCl_3_ (10%) was added after another 5 min, and the reaction mixture was then treated with 0.2 mL of 1 mM NaOH. Finally, the mixture was diluted using water to obtain a total volume of 1 mL, and the absorbance was measured at 510 nm using the ELISA spectrophotometer (uQuant). Rutin (1.5–200 µg/mL) was used as the standard to construct a calibration curve.

#### 4.4.3. Diphenylpicrylhydrazyl Radical Scavenging Activity

Diphenylpicrylhydrazyl (DPPH) radical scavenging activity was measured using the method that was reported by Gyamfi [41], with slight modifications. The reaction mixture included 0.4 mL of 0.5 mM DPPH in EtOH, 0.5 mL of 100 mM Tris-HCl buffer (pH = 7.4), and 100 µL of different concentrations of SBR extract (1–100 µg/mL), or pure compound (baicalin, baicalein, wogonin, and oroxylin A; 75–100 µg/mL) was isolated from SBR extract. Gallic acid was used as the standard to construct a calibration curve. The mixtures were incubated in the dark at room temperature for 30 min, and the absorbance was measured at 517 nm using the ELISA spectrophotometer (uQuant). Scavenging action (%) was calculated using the following equation:Scavenging action (%) = [1 − (absorbance intensity of compound or SBR extract/absorbance intensity of control)] × 100%

For the extracts, the values are reported as mean ± SD of five measurements; half-maximal inhibitory concentration (IC_50_) was calculated using linear regression analysis, and IC_50_ is expressed as the mean of three measurements. Gallic acid was used as a free radical scavenger reference compound.

#### 4.4.4. Ferric Reducing/antioxidant Power Activity

The method that was reported by Benzie and Szeto [42] was used to measure the ferric-reducing activity of each standard solution, with some modifications. Fresh ferric reducing/antioxidant power (FRAP) reagent was prepared by mixing 0.3 M acetate buffer (pH 3.6), 5 mM 2, 4, 6-Tris (2-pyridyl)-s-triazine (TPTZ), and 20 mM ferric chloride (10:1:1, *v*/*v*/*v*). Subsequently, 50 µL of the SBR extract or baicalin, baicalein, wogonin, and oroxylin A (10 µg/mL) was added to 1450 µL of FRAP reagent. After 8 min, readings at the absorption maximum (593 nm) were taken every 15 s using the ELISA spectrophotometer (uQuant). In this assay, excess Fe^3+^ was used, and the reducing ability Fe^2+^ of the sample is the rate-limiting factor of Fe -TPTZ, and hence color formation. Trolox (0.2–30 mM) was used as the positive control to construct a calibration curve. The data was also expressed as Trolox equivalent antioxidant capacity (TEAC) milligrams per gram of sample weight.

#### 4.4.5. Superoxide Anion Radical (O_2_•−) Scavenging Activity

The method that was described by Robak and Gryglewski was used to measure the O_2_•− scavenging activity of each SBR extract [43]. All of the reagents were prepared in 0.2 M sodium phosphate buffer (pH 7.4). Standard compounds (SOD, 1.5–200 U/mL) were used as the reference standard to construct a calibration curve. The SBR extracts or baicalin, baicalein, wogonin, and oroxylin A were serially diluted to obtain different concentrations of test samples (0.75–100 μg/mL SBR extracts or 1.5–200 μg/mL baicalin, baicalein, wogonin, and oroxylin A); 100 μL of the sample solution was then added to 250 μL PMS (120 μM) and 120 μL NBT (300 μM), followed by vigorous mixing. This mixture was then incubated at 37°C in an oven for 5 min; thereafter, 936 μM β-NADH was added. After 10 min, absorbance was measured at 560 nm using the ELISA spectrophotometer (uQuant) against a blank; a low absorbance value indicated increased O_2_•− scavenging activity. The scavenging effect (%) (i.e., the activity to scavenge O_2_•−) was calculated using the following equation: Scavenging effect (%) = [1 − (absorbance of sample at 560 nm/absorbance of control at 560 nm)] × 100%

#### 4.4.6. Hydroxyl Radical Scavenging Activity

Hydroxyl radical (−OH) scavenging activity was measured according to the method that was reported by Halliwell [44], with slight modifications. Various concentrations of SBR extracts (0.8–80 µg/mL), baicalin, baicalein, wogonin, and oroxylin A (0.2–20 µg/mL), or the standard (EtOH, 0.15–20 µM) were added to a reaction mixture (200 µL) containing 30 mM KH_2_PO_4_–KOH buffer (pH 7.4), 2 mM 2-deoxyribose, 0.1 mM FeCl_3_•6H_2_O, 104 µM EDTA, 1.0 mM H_2_O_2_, and 0.1 mM L-ascorbic acid. The solutions were then incubated for 50 min at 37 °C; thereafter, ice-cold TCA (0.5 mL, 2.8%, *w/v*) and TBA (0.5 mL, 1%, *w*/*v*) in H_2_O were added. The reaction mixture was kept in a boiling water bath for 30 min and cooled, and the absorbance was then measured at 532 nm. The scavenging effect (%) was calculated using the following equation: Scavenging effect (%) = 1 – [(absorbance of sample − absorbance of blank)/ (absorbance of sample − absorbance of control)] × 100%

#### 4.4.7. H_2_O_2_ Scavenging Activity

The H_2_O_2_ scavenging ability of the test and standard solutions was measured using the method that was reported by Okuda et al. [45], with some modifications. Various concentrations of catalase (0.04–5 U/mL) were used as the standard to construct a calibration curve. To 0.2 mL of H_2_O_2_ (4 mM in H_2_O), 0.5 mL of the SBR extract (1 mg/mL) or baicalin, baicalein, wogonin, and oroxylin A (100 µg/mL) was added, followed by incubation for 20 min at room temperature; subsequently, 0.3 mL of HRPase–phenol red (0.5 mg/mL HRPase and 7.5 mM phenol red) solution was added. The mixture was incubated at 25 °C for 10 min and transferred to a cold water bath to terminate the reaction. The absorbance was measured at 610 nm using the ELISA spectrophotometer (uQuant). The scavenging effect of H_2_O_2_ (%) was calculated using the following equation:H_2_O_2_ scavenging effect (%) = [1 − (absorbance of sample at 610 nm/absorbance of control at 610 nm)] × 100%

#### 4.4.8. Ferrous Ion-chelating Activity

The method reported by Dinis et al. was used to determine the chelation of ferrous ions [46]. Various concentrations of 20 mg/mL EDTA (0.02–0.25 mg/mL) were used as the standard to construct a calibration curve. For measurement, 0.25 mL of the SBR extract (1 mg/mL) or baicalin, baicalein, wogonin, and oroxylin A (100 µg/mL) was added to 0.7 mL of the reagent (0.675 mL MeOH and 0.025 mL of 2 mM FeCl_2_ in H_2_O). After 30 s, 0.05 mL of 5 mM ferrozine water solution was added, and the reaction was allowed to proceed for 10 min Absorbance was measured at 562 nm using the ELISA spectrophotometer (uQuant). The chelating effect (%) was calculated using the following equation: Chelating effect (%) = [1 − (absorbance of sample at 562 nm/absorbance of control at 562 nm)] × 100%

### 4.5. Determining the Effect of Lipid Peroxidative Product of SBR Extracts on Rat Lung Tissues

#### 4.5.1. Preparation of Rat Lung Mitochondria

The lungs were obtained from male Wistar rats immediately after the animals were sacrificed; the lungs were kept on ice, minced to small pieces, and then washed with saline. The minced lungs were put with ice-cold PBS and homogenized in a Potter Elvehjem homogenizer. The homogenate was then centrifuged at 2000 rpm for 10 min at 4 °C to separate the nuclear debris. The clear suspensions were transferred to a new tube and centrifuged again at 12000 rpm for 10 min at 4 °C to obtain the mitochondrial fraction.

#### 4.5.2. Protein Content Determination

The method that was described by Lowry et al. was used to determine the protein content in the mitochondrial fraction [47]. Bovine serum albumin was used as the standard to construct a calibration curve, and absorbance was measured at 595 nm using the ELISA spectrophotometer (uQuant).

#### 4.5.3. Synthesis of Linoleic Acid Hydroperoxide

Linoleic acid hydroperoxide (LHP) was enzymatically synthesized by the reaction of linoleic acid with soybean lipoxygenase, according to the method that was described by Ohkawa et al. [48], with some modifications. Briefly, 50 mg of linoleic acid was dissolved in 0.1 M borate buffer (pH 9). Subsequently, 15 munit/mg soybean lipoxygenase was added, followed by incubation under saturated oxygen conditions at 4 °C for 2 h. After incubation, 2.5 g of NaCl was added to stop the reaction, and the hydroperoxides were extracted with anhydrous diethyl ether. To dry the ether extract, anhydrous sodium sulfate was used to remove any water, and the extract was filtered. The filtrate was evaporated to dryness in a rotary evaporator, resuspended in MeOH, and stored at −80°C. The maximal absorption at 234 nm was used to calculate the final concentration of LHP by applying Beer’s law while using ε = 23,000 as the molar extinction coefficient. The LHP structure was determined on the basis of its mass (Micromass Quattro LC) and nuclear magnetic resonance (NMR, Varian Unity INOVA 500 MHz spectrometer).

#### 4.5.4. LHP-induced LPO Inhibition Assay

The LPO-inhibitory activity of each compound was determined using the TBA-reactive substance (TBARS) assay [49]. The concentration of malondialdehyde, a compound produced during LPO, was determined using the TBA method. The total reaction mixture volume was 0.25 mL, which contained 0.1 mL of lung mitochondria, 0.1 mL of the inducer (LHP, 40 nmol/mg of protein), and 0.1 mL of the SBR extract (2–200 μg/mL) or a standard compound, namely Trolox (2–1000 μM). The mixture was incubated at 37 °C in an incubator for 60 min The reaction was stopped by adding 0.375 mL of H_3_PO_4_ (44 mM), 0.2 mL of distilled water, and 0.125 mL of TBA (42 mM). The mixture was then incubated at 90b°C in an incubator for 60 min After cooling and centrifugation at 3000 rpm/min for 10 min, the absorbance of the supernatant was measured at 532 nm. The inhibition of TBARS formation (% inhibition of LPO) was calculated, as follows: Inhibition of LPO (%) = (absorbance of sample at 532 nm − absorbance of control at 532 nm)/ (absorbance of control at 532 nm − absorbance of blank) × 100%

### 4.6. Phytochemical Isolation from the EtOAc Extract

The herbs of SBR (l kg) were powdered to a fine grade, immersed in, and twice extracted with, 10-fold v/w of EtOAc at room temperature for two weeks. After concentrating the EtOAc filtrate under reduced pressure to obtain the EtOAc extract, the residues were twice reflux-extracted with four-fold *v/w* of 50% aqueous EtOH for 6 h. Aqueous EtOH (50%) extracts were concentrated to obtain a yellow precipitate. These were recrystallized with aqueous EtOH to obtain compound I. EtOAc extracts were subjected to column chromatography on a silica gel eluted with CHCl_3_–CHCl_3_-MeOH and re-chromatographed on silica gel eluted with CH_2_Cl_2_–acetone to yield compounds II and III. A portion of the CHCl_3_-MeOH elute was subjected to a Sephadex LH-20 column eluted with MeOH to yield compound IV. The identity of the compounds was determined on the basis of mass and NMR (Varian Unity INOVA 500 MHz spectrometer).

### 4.7. Phytochemical Analyses

HPLC was used to identify the phytochemical components in each SBR extract. The HPLC system consisted of a Shimadzu model LC-10AT (Kyoto, Japan) that was equipped with a Shimadzu Model SIL-9A autoinjector and a Shimadzu Model SPD-10 A detector (Shimadzu, Kyoto, Japan). The peak areas were calculated with a Shimadzu Model C-R8A recorder. A LiChrospher 100 RP-18e reversed-phase column (Merck, Darmstadt, Germany) and LiChrospher 100 RP-18e guard column (Merck) were used. The purity of each compound exceeded 98.0%.

### 4.8. Data Analysis

Experiments were performed in triplicates, and results are expressed as mean ± SD.

## 5. Conclusions

Air pollution, particularly PM2.5 exposure, is responsible for increasing the risk of respiratory and cardiac diseases globally. Accordingly, novel nutritional supplements and therapies are needed now more than ever. In this study, we observed that the EtOAc extract of SBR showed high ROS scavenging ability. Moreover, baicalein showed the most significant LHP-induced LPO inhibition in rat lung mitochondria; the EtOAc extract contained the highest amount of baicalein. We conclude that baicalein and the EtOAc extract of SBR can be used as effective agents for lung damage protection. Additional studies are warranted to investigate their use as antioxidant therapy for respiration infections, nutrition supplements, and lead compounds in pharmaceuticals.

## Figures and Tables

**Figure 1 molecules-24-02143-f001:**
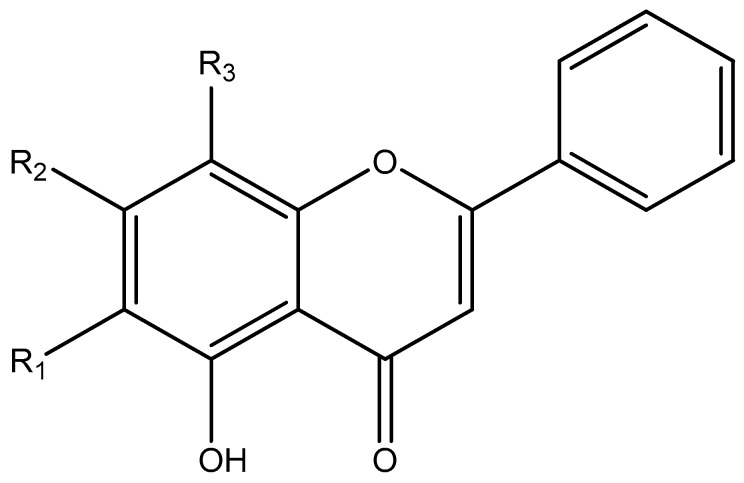
Structure, molecular weight and appearance of (1) baicalin, (2) baicalein, (3) wogonin, and (4) oroxylin A.

**Figure 2 molecules-24-02143-f002:**
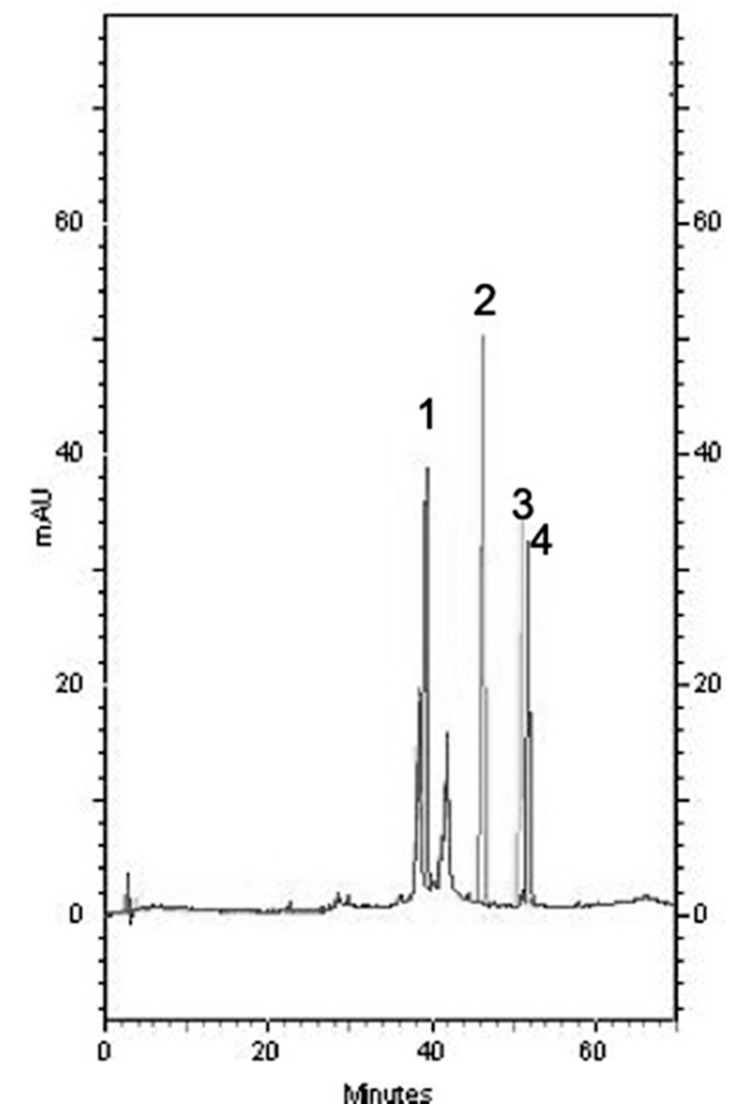
Chromatograph of (1) Baicalin, (2) Baicalein, (3) Wogonin, and (4) Oroxylin A.

**Figure 3 molecules-24-02143-f003:**
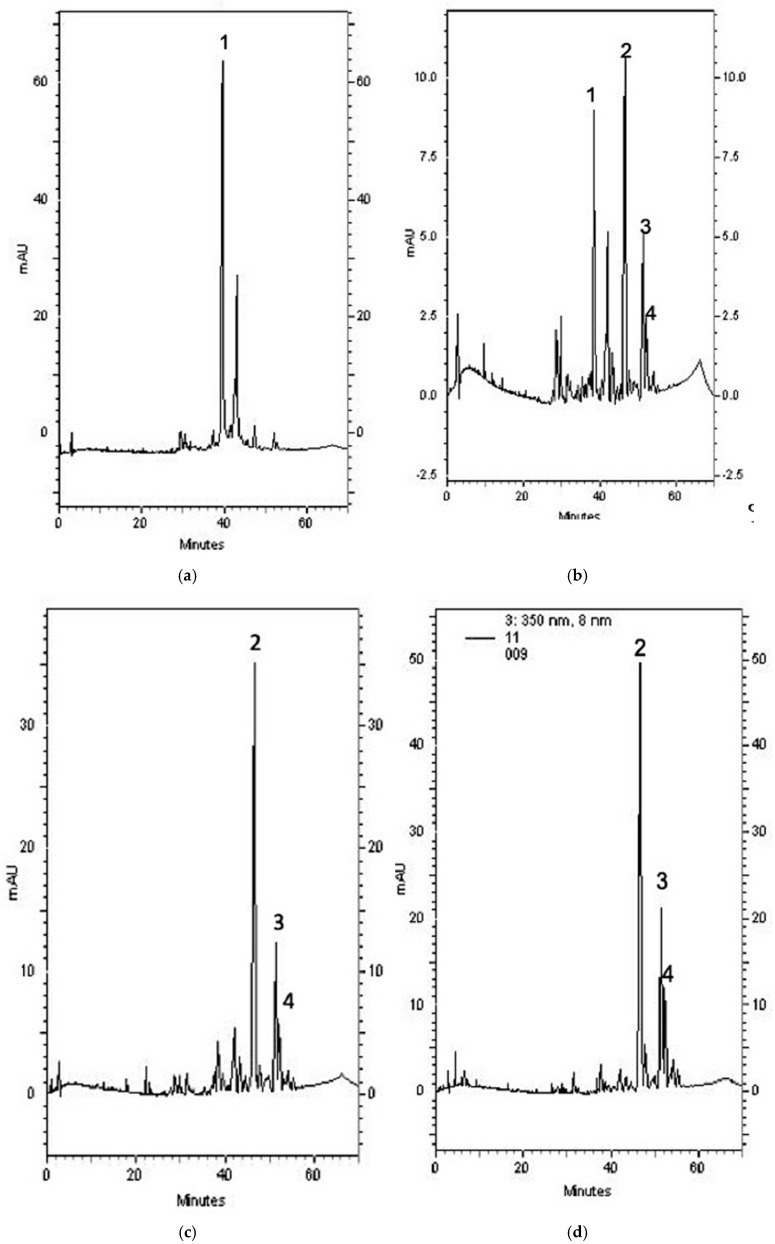
Chromatographs of (**a**) water extract, (**b**) ethanol extract, (**c**) acetone extract, and (**d**) ethyl acetate extract. Peak (1) represents baicalin, (2) represents baicalein, (3) represents wogonin, and (4) represents oroxylin A.

**Table 1 molecules-24-02143-t001:** Effects of *S. baicalensis* root (SBR) extracts on free radical scavenging and antioxidant activities.

In vitro Antioxidant Activity	SBR Extracts of Different Solvent			
	Water	EtOH	Acetone	EtOAc
Total phenol content (µg/mg dry weight)	160.29 ± 6.82	252.81 ± 3.27	284.98 ± 14.93	518.80 ± 34.23
Total flavonoid content (µg/mg dry weight)	884.92 ± 5.72	1076.46 ± 13.31	999.21 ± 13.84	1903.44 ± 69.41
DPPH scavenging activity (IC_50_, µg/mL)	18.17 ± 0.79	24.89 ± 0.33	24.31 ± 0.72	13.08 ± 0.44
Total antioxidant power (TEAC µM on 50 µg/mL)	16.87 ± 0.68	10.37 ± 0.24	19.31 ± 0.44	23.44 ± 0.83
O_2_•− scavenging activity (IC_50_, µg/mL)	81.78 ± 5.33	138.23 ± 12.22	95.46 ± 5.06	66.05 ± 5.11
−OH scavenging activity (IC_50_, µg/mL))	2.92 ± 0.03	2.77 ± 0.07	3.74 ± 0.03	3.11 ± 0.03
H_2_O_2_ scavenging activity (rate% on 1 mg/mL)	48.73 ± 2.57	31.00 ± 1.17	41.73 ± 3.34	47.00 ± 3.50
Fe^2+^ chelating activity (rate% on 1 mg/mL)	78.75 ± 0.73	44.35 ± 2.14	13.10 ± 0.91	1.92 ± 2.90

Data derived from three independent experiments were statistically analyzed, and results are presented as mean ± SD (*n* = 3). Each separate assay was performed in triplicates.

**Table 2 molecules-24-02143-t002:** Inhibitory effect of SBR extracts and baicalein content on linoleic acid hydroperoxide (LHP)-induced lipid peroxidation (LPO) in rat lung mitochondria.

SBR Extracts	Inhibit Rate (%) (200 µg /mL)	IC_50_ (µg /mL)	Baicalein Content (µg /mL)
Water	52.50	24.00 ± 3.87	13.31 ± 0.96
Ethanol	59.17	24.71 ± 4.02	62.29 ± 1.96
Acetone	55.00	2.95 ± 5.28	100.77 ± 1.71
Ethyl acetate	72.50	1.71 ± 5.26	248.05 ± 32.11
Positive control			
Trolox	76.67	20.12 ± 1.44	-

Data derived from three independent experiments were statistically analyzed, and the results are presented as mean ± SD (*n* = 3).

**Table 3 molecules-24-02143-t003:** Inhibitory effects of baicalin, baicalein, wogonin, and oroxylin A on linoleic acid hydroperoxide (LHP)-induced lipid peroxidation (LPO) in rat lung mitochondria.

Compounds	IC_50_ (µg /mL)
Baicalin	6.84 ± 0.11
Baicalein	0.20 ± 0.04
Wogonin	3.30 ± 0.08
Oroxylin A	105.82 ± 0.80
Positive control	
Trolox	20.12 ± 1.44

**Table 4 molecules-24-02143-t004:** Effects of baicalin, baicalein, wogonin, and oroxylin A on free radical scavenging and antioxidant activities.

In vitro Antioxidant Activity	Pure Compound Isolated from SBR			
	Baicalin	Baicalein	Wogonin	Oroxylin A
DPPH scavenging activity (IC_50_, µg/mL))	6.93±0.08	2.80 ± 0.05	>100	>100
Antioxidant power ability (TEAC µM on 30 µg/mL)	19.49±0.57	10.18 ± 0.30	0.43 ± 0.09	0.28 ± 0.03
O_2_•− scavenging activity (IC_50_, µg/mL)	>100	43.99 ± 1.66	>100	>100
−OH scavenging activity (IC_50_, µg/mL))	0.69 ± 0.02	>20	>20	>20
H_2_O_2_ scavenging activity (rate% on 100 µg/mL)	14.67 ± 1.39	11.69 ± 1.29	0	0
Fe^2+^ chelating activity (rate% on 100 µg /mL)	20.07 ± 1.07	2.38 ± 0.69	2.32 ± 0.71	1.32 ± 0.09

Data derived from three independent experiments were statistically analyzed, and results are presented as mean ± SD (*n* = 3). Each separate assay was performed in triplicates.

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
