# Peer review of "Inhibitory Effects of *Scutellaria baicalensis* Root Extract on Linoleic Acid Hydroperoxide-induced Lung Mitochondrial Lipid Peroxidation and Antioxidant Activities"

_molecules, 2019, doi:10.3390/molecules24112143_

Reviewer 1 Report

The topic of the paper is interesting, because it gives some novel information about the potential oxygen free radical scavenging and antioxidant capacity of the Scutellaria baicalensis Root extract made by different solvents.

Lung mitochondria as model is useful and resulted impressive data, which may be use for further, more detailed pharmacological investigations.

The paper is well written and its structure is correct, but requires some editorial modifications for elimination of typing errors. The paper contains lot of abbreviations which requires a list of abbreviations.

There are the following proposals in order of appearance in the text:

r. 14: PM2.5 – this term should explain shortly at first, because not all of the readers are familiar with it.

r. 18: free radicals are cause and LPO is consequence of them (please re-phrase the sentence)

r. 62:  Please modify the sentence because it is not clear that four different extraction was made by different solvents and the extraction did not make with four solvents.

r. 64: Please add the storage temperature (there are different refrigerators)

Table 1 (and later): some dimensions are not clear, e.g. mmM on 50 g/ml etc.) Please explain them in the footnote of the Table

r. 76-78: The difference means statistical difference, therefore if the difference was statistically not proven should be mention more carefully. In this case the statement is only hypothetical, even if the difference numerically seems to be relevant.

r. 165: please mention here that hydroperoxide is not a per se free radical

r. 179-180: Chelating the trace elements is a really important process, but it is not true in respect of ROS formation for all of them, only for transient metals

r. 194-195: This sentence should re-phrase because it suggest that malondialdehyde is the causative agent for cell injury.

r. 231-232:  see my comment for r. 62

r. 237: The term ‘perfuse’ not correct, because minced samples could not perfuse (maybe wash)

r. 365: Please add the concentration of phosphoric acid and TBA

r. 388-389: Title of the subchapter is not correct, because there was no real statistical evaluation of data only calculate the means and standard deviation.

Author Response

Thank you for editing the manuscript molecule -514754 and providing us with some valuable feedback. Based on your comments, we made the following changes:

We corrected the content (symbols, grammar, sentence patterns, etc.) as suggested and added abbreviations in the manuscript.

Point 1: The paper is well written and its structure is correct, but requires some editorial modifications for elimination of typing errors. The paper contains lot of abbreviations which requires a list of abbreviations.

Response 1: Thank you for your kind suggestion. We have revised some errors and have made a list of abbreviations.

Point 2: r.14: PM2.5 – this term should explain shortly at first, because not all of the readers are familiar with it.

Response 2: Thank you for kind suggestion.  Because there is no experiment directly related to PM2.5 in the manuscript, the contents of PM2.5 were thus deleted according to other reviewer suggestions.

Point 3: r.18: free radicals are cause and LPO is consequence of them (please re-phrase the sentence)

Response 3: Thank you for your constructive suggestions. The sentence has been corrected to be “ In this study, we evaluated the ability of Scutellaria baicalensis Georgi to protect lipid-peroxidation (LPO) in lung tissue after free radical-induced injury.”

Point 4: r. 62: Please modify the sentence because it is not clear that four different extraction was made by different solvents and the extraction did not make with four solvents.

Response 4: We agree with the reviewer and have added the following sentences in the result (lines 67- 68): SBR was extracted with water, ethanol (EtOH), acetone, and ethyl acetate (EtOAc) to yield four extracts.

Point 5: r. 64: Please add the storage temperature (there are different refrigerators).

Response 5: Thanks for your reminder. We added the storage temperature (-20°C) at line 69.

Point 6:Table 1 (and later): some dimensions are not clear, e.g. mmM on 50 g/ml etc.) Please explain them in the footnote of the Table

Response 6: Thank you for this reminder. The dimensions of Tables have been supplemented.

Point 7: r. 76-78: The difference means statistical difference, therefore if the difference was statistically not proven should be mention more carefully. In this case the statement is only hypothetical, even if the difference numerically seems to be relevant.

Response 7: We agree and thank you for your reminder.

Point 8: r. 165: please mention here that hydroperoxide is not a per se free radical

Response 8: Thank you for this suggestion and we have added the following sentences in the discussion (line 166-167): “H2O2 would be catalyzed by enzyme to produce hydroxyl free radical –OH which is very reactive and rapidly attack the molecules in nearby cells…”

Point 9: r. 179-180: Chelating the trace elements is a really important process, but it is not true in respect of ROS formation for all of them, only for transient metals

Response 9: We agree and thank you for your reminder. So we will modify the sentence as follows: Suppressing ROS formation either by chelating transition metals or inhibition of enzymes involved.  (line 182)

Point 10: r. 194-195: This sentence should re-phrase because it suggests that malondialdehyde is the causative agent for cell injury.

Response 10: Thank you for this suggestion. We fixed the sentences shown at lines 197-199.

Point 11: r. 231-232: see my comment for r. 62

Response 11: We agree and thank you for your reminder.

Point 12: r. 237: The term ‘perfuse’ not correct, because minced samples could not perfuse (maybe wash)

Response 12: Thank you for this suggestion. We correct the term “perfuse” to “put” (line 340). 

Point 13: r. 365: Please add the concentration of phosphoric acid and TBA

Response 13: Thank you for this reminder. We have added the concentration of phosphoric acid and TBA (line 368)

Point 14: r. 388-389: Title of the subchapter is not correct, because there was no real statistical evaluation of data only calculate the means and standard deviation.

Response 14: We appreciate your reminder. We have changed the title of " Statistical analyses " to "Data analysis"

Reviewer 2 Report

The manuscript by Liau et al., presents an interesting topic, reporting on the investigation of antioxidant activity of Scutellaria baicalensis roots extracts and their ability to inhibit lipid peroxidation in rat lungs. This is a matter of interest because the oxidant/antioxidant balance play an important role in lung dysfunction and this kind of extracts could be considered as antioxidant therapy or nutrition supplement.

The review of the literature presented in the introduction is, in general, adequate. However, since the authors start talking about PM2.5, I suggest to discuss in more detail the link between PM.2.5 and oxidative stress.

There are some points that could be improved:

-          The Folin-Ciocalteu (FC) method is commonly used to determine the total phenolic content (TPC) of natural extract, it is not specific for phenolics, since the FC reagent can react with other substances present in plant extracts (e.g., ascorbic acid, reducing sugars…) causing an overestimation of TPC. Can the authors exclude the presence of this kind of interference? Please, at least discuss this issue.

-          Line 79: “Moreover, antioxidant and −OH and H2O2 scavenging activities were not-directly associated with total flavonoid and phenol content.” Did the author try to find a correlation between total phenol/flavonoid content and antioxidant activity? Can the authors suggest  what the antioxidant properties could be ascribed to?

-          2.2. Total flavonoid and total phenol contents of SBR extracts: I suggest to move this paragraph up, since the total phenol and flavonoid content is recalled in the previous paragraph.

-          The four compound identified in the EtOH belong to the flavone group, a class of flavonoid, but the author stated  that the antioxidant activity is not associated with the flavonoid content. The authors could discuss what looks like an inconsistency.

-          Spectrophotometer assays: did the authors check that the samples did not absorb at the wavelength used for the assay and possibly correct the contribution due to the color of the extract?

-          Line 382: the authors stated that they have used HPLC to identify phytochemical components in each SBR extract. Did they use any standard for identification?

.

Some minor mistakes or inconsistencies:

-          Abstract: please explain the acronyms ROS and LPO the first time they appear in the text.

-          Table 1: um should be mg.

Author Response

Thank you for editing the manuscript molecule -514754 and providing us with some valuable feedback. Based on your comments, we made the following changes:

We corrected the content (symbols, grammar, sentence patterns, etc.) in the manuscript as suggested.

Point 1: The Folin-Ciocalteu (FC) method is commonly used to determine the total phenolic content (TPC) of natural extract, it is not specific for phenolics, since the FC reagent can react with other substances present in plant extracts (e.g., ascorbic acid, reducing sugars…) causing an overestimation of TPC. Can the authors exclude the presence of this kind of interference? Please, at least discuss this issue.

Response 1: We concur and thank you for this reminder. In general, ascorbic acid and reducing sugars belong to polar compounds, and won’t be extracted easily by organic solvent (ethanol, acetone, and ethyl acetate). So, the data of total phenol content would not be interfered with ascorbic acid and reducing sugars.

Point 2: Line 79: “Moreover, antioxidant and −OH and H2O2 scavenging activities were not-directly associated with total flavonoid and phenol content.” Did the author try to find a correlation between total phenol/flavonoid content and antioxidant activity? Can the authors suggest what the antioxidant properties could be ascribed to?

Response 2: Thank you for your suggestion. Because the role of antioxidants has different mechanisms, such as anti-free radicals, affecting the oxidation potential of H2O2, so there are many substances that can affect free radical reactions, some will chelate with metals, and some It will reduce the reaction of OH-. Therefore, the scavenging activities of antioxidants and -OH and H2O2 are not directly related to the total flavonoids and phenol content. Therefore, we have carried out various antioxidant-related activity screenings. The mechanism of flavonoids and phenols is clearer related to free radical reactions or chelation with metals, but different substituents can have different effects.

Point 3: 2.2 Total flavonoid and total phenol contents of SBR extracts: I suggest to move this paragraph up, since the total phenol and flavonoid content is recalled in the previous paragraph.

Response 3: Thank you for this suggestion. We have moved the paragraph “Total flavonoid and total phenol contents of SBR extracts” from section 2.2 to 2.1 (line 60).

Point 4: The four compounds identified in the EtOH belong to the flavone group, a class of flavonoid, but the author stated that the antioxidant activity is not associated with the flavonoid content. The authors could discuss what looks like an inconsistency.

Response 4:  Thank you for your suggestions. Because the antioxidant activities of antioxidants have different reaction mechanisms, some carry out free radical reactions, some are metal chelate reactions, etc., flavonoids will be chelated with metals but not with H2O2. Therefore, antioxidant activity and flavonoid content may not have a positive relationship.

Point 5: Spectrophotometer assays: did the authors check that the samples did not absorb at the wavelength used for the assay and possibly correct the contribution due to the color of the extract?

Response 5: Thank you for this reminder. Before the spectrophotometer assays, we had checked the absorbance of the sample were not interfered at the wavelength.

Point 6: Line 382: the authors stated that they have used HPLC to identify phytochemical components in each SBR extract. Did they use any standard for identification?

Response 6: Thank you for this reminder. We used commercial reference standard to identify phytochemical components, and the chromatogram of reference standards has been shown in figure 2.

Point 7: Some minor mistakes or inconsistencies:

[1]      Abstract: please explain the acronyms ROS and LPO the first time they appear in the text.

Response 7: Thank you for this suggestion.  We have added original words in the abstract.

[2]      Table 1: um should be mg.

Response 7: Thank you for this suggestion.  We have corrected it.

Reviewer 3 Report

The authors no evaluated the effect of PM2.5, this idea must remove the document, because there did no reported evidence of the evaluation of this concept (PM2.5)

p.p1 {margin: 0.0px 0.0px 0.0px 0.0px; font: 13.0px Helvetica}

The introduction must be replanted and the references employed in this proposal must be adjusted in order to the new orientation. That it must be according with the describe objective approached ...."S. baicalensis root (SBR) extracts to protect the lungs from damage caused by 58 lipid peroxidation (LPO) and free radicals". 

Author Response

Thank you for editing the manuscript molecule -514754 and providing us with some valuable feedback. Based on your comments, we made the following changes:

We corrected the content (symbols, grammar, sentence patterns, etc.) in the manuscript as suggested.

Point: The authors no evaluated the effect of PM2.5, this idea must remove the document, because there did no reported evidence of the evaluation of this concept (PM2.5). The introduction must be replanted and the references employed in this proposal must be adjusted in order to the new orientation. That it must be according with the describe objective approached ...."S. baicalensis root (SBR) extracts to protect the lungs from damage caused by 58 lipid peroxidation (LPO) and free radicals".

Response : Thanks for this valuable suggestion. We have removed the contents of PM2.5 and rewritten the introduction in accordance with the objective.